# Mitigation of Calcium-Related Disorders in Soilless Production Systems

**Virginia Birlanga** [1] , **José Ramón Acosta-Motos** [2,3,*] and **José Manuel Pérez-Pérez** [4,*]

1   Bayer CropScience, 30319 Miranda, Spain; virginia.birlanga@bayer.com
2   Group of Fruit Tree Biotechnology, CEBAS-CSIC, 30100 Murcia, Spain
3   Cátedra Emprendimiento en el Ámbito Agroalimentario, Campus de los Jerónimos, Universidad Católica San Antonio de Murcia (UCAM), no. 135 Guadalupe, 30107 Murcia, Spain
4   Instituto de Bioingeniería, Universidad Miguel Hernández, 03202 Elche, Spain
*   Correspondence: jracosta@ucam.edu (J.R.A.-M.); jmperez@umh.es (J.M.P.-P.);
    Tel.: +34-968-278-800 (J.R.A.-M.); +34-966-658-958 (J.M.P.-P.)

**Abstract:** In the current scenario of human-driven climate change, extreme weather events will likely affect agricultural production worldwide. Soilless production systems have recently arisen as a solution to optimize the use of natural resources, such as water and soil, and hence will contribute to reducing the environmental impact of agriculture. However, nutritional imbalance due to adverse environmental factors, such as drought, high temperatures, and salinity, might produce calcium-related physiological disorders during plant growth, such as blossom-end rot (BER) in fruits and tipburn (TB) in leaves, which are a serious problem in crop production. Here, we discuss the different agronomic, physiological, and genetic factors that favor the induction of BER in tomato and TB in lettuce and anticipate the use of an integration of breeding and technological approaches to alleviate nutritional disorders in soilless production systems.

**Keywords:** climate change; soilless agriculture; blossom-end rot; tipburn; calcium deficiency

## 1. Climate Change and Agriculture

Scientists now agree that human activities are the main drivers of climate change [1]. Agriculture, forestry, and other land uses contribute approximately 13% of the carbon dioxide ($CO_2$), 44% of the methane, and 81% of the nitrogen oxide emissions, which together represent 23% of the net greenhouse gas (GHG) emissions [1]. However, only 29% of total anthropogenic $CO_2$ emissions during the 2007–2016 period were neutralized by Earth's natural responses; hence, it is expected that the global atmospheric $CO_2$ levels will further increase [2].

Human-driven climate change will enhance extreme weather events that negatively affect terrestrial ecosystems. As a result, there will be an increase in degradation and desertification in many regions of the planet, leading to a reduction in crop yields and consequently food security will be affected globally. Soil is both a source and a sink for GHGs and performs a crucial role in the exchange of energy, water, and aerosols between the soil surface and the atmosphere. Therefore, sustainable soil management can help mitigate the negative impacts of various environmental stressors, especially those dependent on climate change, on ecosystems, and societies [2]. Farmers are now implementing a set of agricultural practices to reduce the effects of climate change, through changes in tillage practices, the selection of crop species and cultivars that grow more efficiently and are better adapted to adverse conditions, as well as through the implementation of a more sustainable use of natural resources. Therefore, a proper balance must be found by considering the contributions of these new practices to produce better yields, increasing farmers' incomes and other environmental indicators [3,4].



The objective of this review is to provide farmers with a compendium of strategies, tools, and solutions to problems directly related to crop quality, such as blossom end rot (BER) on fruits and tipburn (TB) in leaves. In this study we will analyze the possible triggers of these physiological disorders by studying the environmental factors directly related to climate change. In addition, new production and managing strategies will be described for a more efficient use of resources that contribute to reducing the appearance of these symptoms, whether due to environmental or genetic factors or a combination of both.

## 2. Soilless Production Systems: Challenges and Solutions

A new model of industrial-scale agriculture, known as soilless agriculture, has emerged in recent years as a system that optimizes the use of natural resources, such as water and soil, and that allows for better environmental control due to its implementation indoors. Soilless agriculture contributes to better plant growth thanks to an adequate management of the root zone in terms of a more uniform and precise control of water and fertilizer needs. With this technique, it is possible to produce healthy vegetables of excellent quality [5,6].

Soilless agriculture not only improves the quality of agricultural products, but also contributes to the reduction of their environmental impact by ensuring a more efficient use of water and fertilizers, mainly nitrates and phosphates ($NO_3^-$ and $PO_4^{2-}$), which can reach rivers and seas due to leaching by torrential rains, causing the contamination of surface waters by eutrophication [7]. The possibilities provided for helping reduce the environmental impact of agricultural systems include the reuse of industrial waste as a growing medium. Soilless cropping systems in which 50% of the drainage was recirculated, reduced $NO_3^-$ and $PO_4^{2-}$ emissions as compared to systems without drainage recovery [8,9]. At present, soilless farming has become consolidated as a suitable tool to optimize intensive crop production and reduce the use of non-renewable resources.

### 2.1. Soilless Cropping Systems

Soilless cropping systems can be classified into several types based on the use of the nutrient solution or the physical state of the root growth media (Table 1). Consequently, we distinguish between open-loop systems (Figure 1a), if the nutrient solution is discarded after use, or closed-loop systems (Figure 1b), if the nutrient solution is reformulated after use and returned to the system. The nutrient solution consists of water, oxygen ($O_2$), and all essential plant nutrients [10]. The root system could grow in the air (aeroponic cultivation), on the liquid nutrient solution (hydroponic cultivation), and on a solid substrate with added nutrient solution (substrate cultivation).

**Table 1.** Classification of soilless cropping systems.

| Classification | Categories | Characteristics |
|---|---|---|
| Nutrient solution use | Open-loop systems | The used nutrient solution is discarded |
| | Closed-loop systems | Nutrient solution is reformulated and returned to the system |
| Physical state of root growth media | Gaseous (aeroponic cultivation) | Spray column system<br>Schwalbach system<br>Aero-Gro system |
| | Liquid (hydroponic cultivation) | Deep floated technique<br>Nutrient film technique<br>New growing system technique |
| | Liquid (aquaponic cultivation) | Nutrient solution is derived from waste from fish production |
| | Solid (substrate cultivation) | Directly in substrate<br>Systems of cultivation in bags or containers<br>Single unit culture systems |

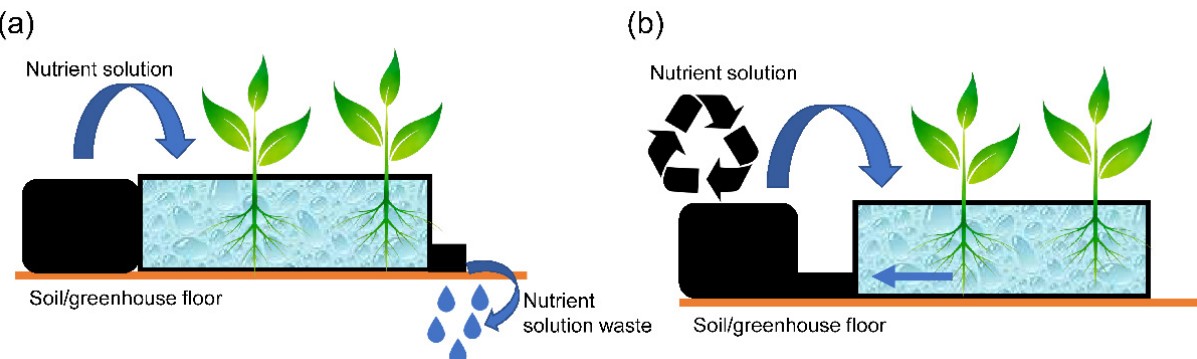

**Figure 1.** Soilless cropping systems as regards to nutrient solution uses. (**a**) A scheme of an open-loop system in which nutrient solution residues are not recycled, and (**b**) a scheme of a closed-loop system in which nutrient solution residues are reintroduced into the system.

Soilless cropping systems have been improved over time, showing many advantages as compared to conventional systems, because they avoid direct contact with the soil and therefore minimize the problems related to soil diseases [11].

In open-loop systems (Figure 1a), allowing an excess of nutrients and water to the plants compensates for irregular transpiration, prevents salt accumulation, and corrects nutritional imbalances. In these systems, however, a large amount of nutrients and water is drained away, thus increasing production costs and contaminating the surrounding environment [11]. In contrast, in closed-loop systems (Figure 1b) the drainage solution is collected onto a reservoir for additional treatments to reduce the risk of root-borne diseases and to reformulate the nutrient composition, which might then be used for other plots or reintroduced into the system.

### 2.1.1. Aeroponic Cultivation

For this type of cultivation, the roots are suspended in the air in dark chambers. The nutrient solution is normally sprayed onto the root system at scheduled intervals for optimal aeration [6].

Some of these systems are as follows:

- Spray column system: This consists of a cylindrical platform made of opaque polyvinyl chloride, with lateral perforations through which the plants are introduced. The nutrient solution is sprayed over the upper part of the roots to ensure a permanent contact with the nutrient solution while the lower part of the root is well aerated (Figure 2a).
- Schwalbach system: This consists of a growth chamber in which the roots grow in the air and are kept in complete darkness. The nutrient solution is sprayed at different distribution points located near the leaves to ensure optimal foliar application, after which it drains to the root, where the excess solution is recovered (Figure 2b).
- Aero-Gro system: The nutrient solution is injected onto the roots directly through finely separated droplets at low pressure, avoiding clogging problems in pipes and spray nozzles (Figure 2c).

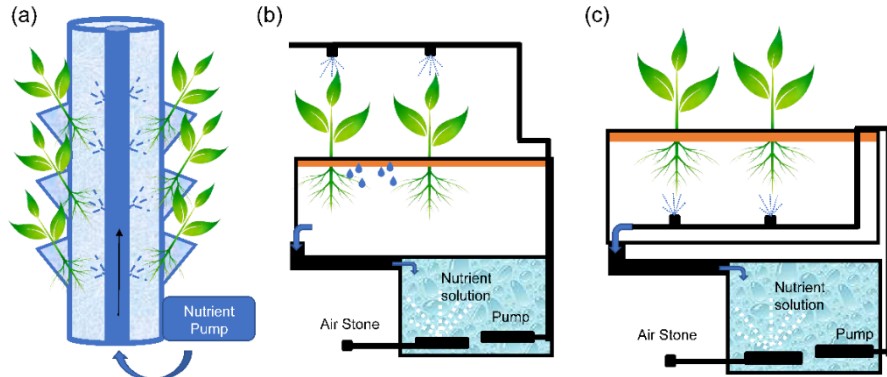

**Figure 2.** Aeroponic cultivation systems. (**a**) Spray column system, (**b**) Schwalbach system, and (**c**) Aero-Gro system.

### 2.1.2. Hydroponic Cultivation

As stated above, in hydroponic cultivation the roots are completely submerged in the nutrient solution without any solid substrate. It is very important for light not to reach the nutrient solution to avoid algal blooms, as this would result in low oxygen availability, and this may affect root growth, and consequently, result in reduced plant yield [11].

There are different types of hydroponic systems:

- Deep floating technique (DFT): It incorporates perforated polystyrene sheets as growing units that are placed on top of the tanks filled with the nutrient solution. The aerial part of the plants grows on these sheets with their roots submerged in the tank solution. These systems have an air pump that aerates the nutrient solution (Figure 3a).

- Nutrient film technique (NFT): This system is based on pumping a thin layer of nutrient solution onto the root system through constant flow. This is achieved by placing a small channel with a 1% slope to ensure that the nutritive solution reaches the roots by laminar flow. The excess solution drains into a collecting tank where the conductivity and pH values are restored and the nutrient solution can be pumped back to the top of the channel (Figure 3b).

- New growing system technique (NGST): This system is based on a channel formed by polyethylene bags located internally in three interconnected layers and wrapped by a layer of black polyethylene, which prevents direct contact of light with the root system. The entire system is suspended in the air and leveled to collect drainage at the end of the growing line. The irrigation system is in continuous operation and the drained solution reaches a tank where the nutrient levels are adjusted, heated, and pumped back into the system. The irrigation pipe is located close the root system to facilitate heating of the roots [11] (Figure 3c).

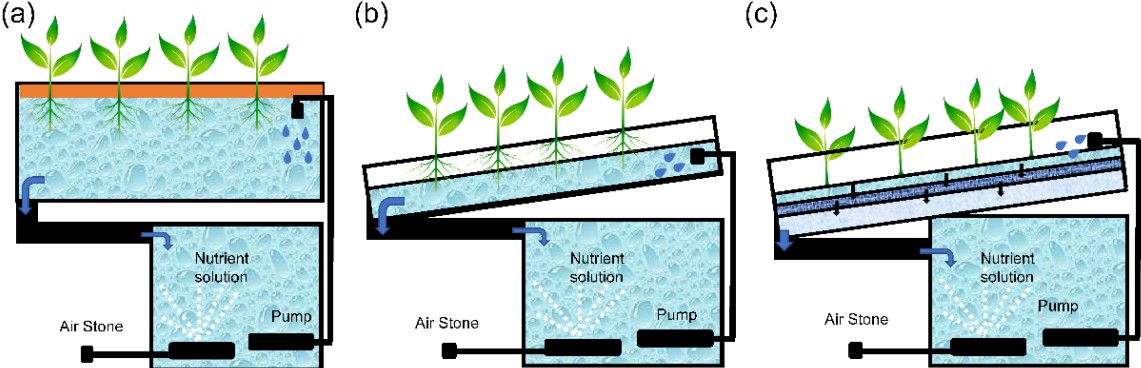

**Figure 3.** Hydroponic cultivation systems. (**a**) Deep floating technique (DFT) system, (**b**) nutrient film technique (NFT) system, and (**c**) new growing system technique (NGST).

### 2.1.3. Aquaponic Cultivation

The concept of aquaponics is based on integrating the industrial production of fish (aquaculture) with the cultivation of plants (horticulture), with the aim of establishing a nutritional balance between both species, in such a way that the use of resources (water and nutrients) is shared in the same production system [12,13]. It is based on the use of waste from the aquaculture production, totally or partially, as a nutrient solution for plant growth in a hydroponic cultivation system (Figure 4) [12–14].

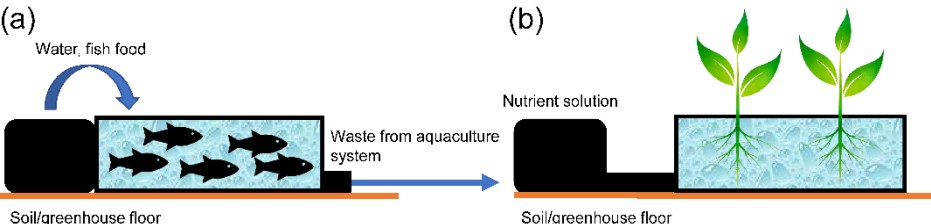

**Figure 4.** Aquaponic system consisting of an (**a**) aquaculture system for fish production, which is connected to a (**b**) hydroponic system used for crop production.

### 2.1.4. Cultivation in Organic and Inorganic Substrates

These systems are based on the use of different substrates that provide optimal oxygen and humidity conditions for the correct development of the plant. Organic substrates of natural origin such as peat, or substrates derived from by-products of agricultural activity, such as coconut fiber, cereal straw, or wood shavings, can be used. Inorganic substrates of natural origin with a high porosity, such as sand or volcanic gravel, can also be applied. In addition, inorganic substrates resulting from the industrial transformation such as rock wool, fiberglass, perlite, or vermiculite, are also frequently used [10]. Substrate cultivation systems are characterized by better aeration as compared to water cultivation systems, but at the same time, the flow of water must be continuous to achieve maximum production [11].

Three systems can be distinguished:

- Growing directly on substrate: These systems are delimited by a thick polyethylene mat that prevents the nutrient solution from leaking into the soil. The irrigation system utilized is drip irrigation, and the excess nutrient solution is sent to a tank where the appropriate adjustments will be made for reusing the nutrient solution (Figure 5a,b).
- Growing in bags or containers: The root volume is delimited by elongated two-color polyethylene bags closed at the ends and with two drainage holes filled with substrate. The plants will grow in these bags, the nutritive solution will be dripped in, and the excess solution will be channeled to a tank for further adjustment and reuse [11] (Figure 5c).
- Single unit culture systems: These systems were developed due to the need to control the transmission of fungal diseases in the continuous systems. In this case, the container is the basic cultivation unit and is placed parallel to the drip line. This allows for better control of individual plants, but this system in large-scale production could be prohibitively expensive (Figure 5d,e).

From an environmental and economic point of view, and with the aim of developing cultivation systems that are as sustainable as possible, the implementation of these new types of soilless cropping systems is increasingly widespread. The improvements in these new production systems allow better usage of the nutritive solution and the reuse of the substrates and other supplies, which will lead to a mitigation of the environmental impact of modern agricultural production [10,11].

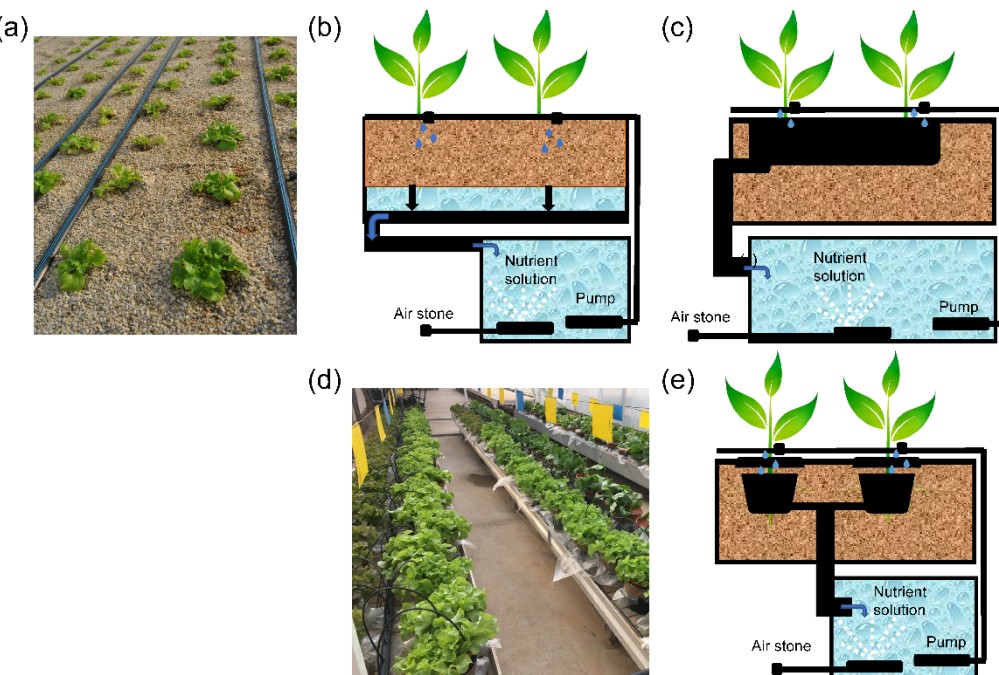

**Figure 5.** Substrate cultivation systems. (**a**) Lettuce plants growing in sand as an inert substrate, (**b**) a scheme of plants growing directly onto substrate, (**c**) a scheme of plants growing in containers, (**d**) lettuce plants growing in individual containers, and (**e**) a scheme of plants growing in individual containers.

### *2.2. Physiological Disorders in Soilless Cropping Systems*

Various physiological disorders can arise in plants growing in these soilless cropping systems and are mainly caused by nutritional imbalance due to adverse environmental factors and not by the effect of the nutrients themselves [15,16]. Environmental stresses such as those related to temperature, irradiation, or relative humidity, favor the appearance of different physiological disorders [16–19]. The most common of these physiological disorders are BER in fruits such as tomatoes, peppers, squash, cucumbers, melons, etc., and TB, which causes necrosis of the leaf margins in leafy crops such as lettuce or cabbage. BER and TB are generally caused by environmental factors such as soil moisture fluctuations, salinity, and heat stress, among others [18,20]. In these cases, endogenous calcium ($Ca^{2+}$) content is reduced, and the rapidly growing tissues are mostly affected. Good management of greenhouse environmental conditions, the use of stress-tolerant varieties, and proper handling of the nutrient solution can alleviate these problems [10,21].

### 3. Physiological Disorders: Blossom-End Rot (BER) and Tipburn (TB)

Climate change is one of the main challenges facing the agricultural sector. Water quantity and quality will be mainly affected by rising temperatures in the coming decades [3,22]. The flexibility of plants to cope with these environmental stresses will depend on their adaptability, and the search for more tolerant genotypes will require the implementation of different strategies to avoid negative effects on plant growth and development.

Physiological disorders such as BER and TB are on the rise due to climate change, are often difficult to predict, and the challenge of controlling the onset of symptoms makes them a serious problem in crop production [15,23,24]. In the early 20th century, BER was believed to be caused by parasitic organisms, chemical toxicity, high transpiration, and lack of soil moisture [25]. However, from the middle of the 20th century, the appearance of BER in tomato, pepper, or watermelon and TB in leafy vegetables, was directly associated to mild $Ca^{2+}$ deficiency in fruits and leaves, respectively [15,18].

Following the new soilless cropping techniques, where the producer provides the necessary nutrient levels that the plant requires at any time, the possibility that mild $Ca^{2+}$

deficiency alone is not the main cause of these physiological disorders is beginning to be assessed [18,26]. Several authors speculate on the cause–effect relationship of $Ca^{2+}$ deficiency in both disorders, observing that on many occasions, fruits with these symptoms contained equal or higher concentrations of $Ca^{2+}$ in their tissues [26]. Other studies indicated that either low levels or high levels of $Ca^{2+}$ in the nutrient solution led to the appearance of BER in fruits of various species [27]. These results suggest that $Ca^{2+}$ deficiency by itself may not be the causative of BER, but rather that a nutrient imbalance is involved in its appearance. Many authors, in their eagerness to predict and act on time against these problems, have centered their attention on the study of the main triggers of these physiological disorders. These studies distinguish different agronomic, physiological, and genetic factors that favor the induction of BER and TB [15,18,20,26,28]. In the following sections, we will succinctly describe all these factors and their relationships.

### 3.1. Abiotic Factors Influencing BER and TB

### 3.1.1. Drought

In agriculture, droughts are generally defined as the periods in which the water losses by transpiration through the leaves, and by evaporation through the soil exceed the amount of water input from precipitation and subsequent water uptake by the roots of the plants. The incidence and intensity of droughts have increased in some regions of the earth, and these are expected to rise in future climate change scenarios [29]. As plants require water for their metabolism, periods of drought can be fatal by reducing crop production to near-unproductive levels (or even causing crop death) or, at best, result in low yields and low-quality products. Depending on the decrease in the irrigation levels by droughts, lettuce and carrot yields are expected to be 25–30% lower than usual. Under these conditions, vegetables and fruits such as apples and pears will generally be sweeter, but smaller. For this reason, consumers and markets will likely have to change their expectations [24].

Studies have been carried out on lettuce and tomato with different irrigation regimes, and under field and greenhouse conditions, to assess commercial traits such as growth, crop maturity, and marketability. Several studies have indicated that deeper roots are key drivers of drought tolerance in plants [30,31]. In addition, a higher incidence of BER and TB has been associated with insufficient water uptake by roots [32–35]. In experiments with different cultivars of lettuce, it was found that TB occurred more frequently in iceberg lettuce than in butterhead lettuce, which is also more drought tolerant [34,36,37]. A similar situation was found in tomatoes, where cultivars with larger fruits suffered more BER symptoms than cultivars with smaller tomatoes [19]. Therefore, breeding new varieties with high drought tolerance is essential for developing vegetables that are better adapted to the consequences of projected climate change, such as BER and TB [36,38].

### 3.1.2. High Temperature

Global warming has led to the increase in the timing, intensity, and duration of heat-related impacts, such as heat waves [2]. In fact, several studies have shown that high temperatures can cause physiological, biochemical, and morphological changes in crops, which leads to inadequate plant development and consequently to yield losses [39–41]. In broccoli, heat can cause malformations, such as uneven heads with large flower buds, bracts on the heads, or soft heads [42]. On the other hand, it has been observed that flavonoid content and glucosinolate composition in the broccoli florets increased with higher temperatures [43].

High light intensities and high temperatures are environmental factors that cause accelerated photosynthesis and growth rates that can trigger BER in tomato and TB in lettuce [42,44–49]. Hence, BER is likely to occur in fruit tissues as the rapid growth rate increases exponentially, and $Ca^{2+}$ supply to other parts of the plant is restricted by mass flow of free $Ca^{2+}$ through the xylem [20]. It has been hypothesized that an increased demand for $Ca^{2+}$ in rapidly growing tissues, such as occurs in fruits during cell growth,

might indirectly lead to BER when $Ca^{2+}$ is limited [50]. In addition, it has been proposed that light and temperature influence $Ca^{2+}$ absorption and distribution within the plant, thus limiting $Ca^{2+}$ concentration within the fruit during heat stress [17]. Lettuce grown at high temperatures display an accelerated growth that causes little uniformity in the closure of the head and enhances early flowering of lateral stems, which causes bitterness in the leaves and enhances TB incidence [16,36,37,49,51].

### 3.1.3. Salinity

Salinity is one of the most recognizable factors that will be enhanced by climate change. In any climate change scenario, salinity occurs because of global warming, which causes ice melting at the poles and thus a rise in sea levels. This further causes coastal waters to come into contact with farmland and contaminate the soil with high levels of salts, which will directly affect the growth, quality, and yield of crops that are grown near the coastline. Saline soils encompass approximately 10% of the land surface, and 50% of irrigated land worldwide [52]. High salt levels in soil lead to aggravated dehydration of plant cells, ion toxicity, and oxidative stress, which can cause growth inhibition, damage at the molecular level, and even plant death [31,53]. In addition, soil salinity prevents nutrient uptake by the plant and alters the permeability of the plasma membrane, causing increased salt accumulation in some plant tissues [17,54]. In fact, the selective uptake of $Ca^{2+}$ over $Na^+$ is a suitable indicator of salinity stress [55]. Alam and co-authors [56] studied the response of 27 tomato genotypes to various salt treatments to determine their response. They observed that the seedlings from the saline treatments had higher concentrations of Na+ in the leaves, as well as greater root length, fresh and dry weight. It has been observed that in soils with a heterogeneous distribution of salts, the root system absorbs significantly more $Ca^{2+}$ than $Na^+$, and this could be a critical factor that contributed to greater $Ca^{2+}/Na^+$ in the fruit and, therefore, to a lower incidence of BER observed in tomato fruits grown in these soils [57]. These authors studied the effect of different saline irrigation regimes under different potential limits of the soil matrix on tomato crop yield and reported BER incidence. The effects of salinity stress on the growth of two types of lettuce under the NFT hydroponic system were analyzed, and it was observed that the amount of fresh and dry matter of the different lettuce types were significantly affected by salinity levels [58].

Water with a low salt content enhanced tomato quality, including fruit density, soluble solids, total acid, vitamin C, and sugar–acid ratio, and had a lower BER incidence than the other more saline treatments. High salinity levels led to a reduction in tomato yield, a decrease in leaf area index and chlorophyll content, together with the appearance of BER symptoms. All this evidence shows that tomato has a moderate salt tolerance index, and mild salinity levels improve osmotic regulation, increase adenosine triphosphatase enzyme activity, and stimulate crop growth [59] Additionally, mild salinity enhanced tomato sensory attributes due to increases in sugar, organic acid, and amino acid contents [59,60]. Inoculation of growth-promoting rhizobacteria in tomato plants has also been shown to improve growth and stress tolerance, resulting in higher crop yields [61–65].

### 3.2. Physiological Factors Influencing BER and TB Incidence

During agricultural production, an appropriate nutrient management is fundamental for the control of BER and TB. It has been observed that when some nutrients, such as K, P, and Mg, are applied above a certain concentration in the nutrient solution (80, 400, and 500 mg $L^{-1}$), they could decrease $Ca^{2+}$ uptake and increase BER incidence [66]. Indeed, reducing $K^+$ supply in combination with the use of fertilizers such as $Ca(NO_3)_2$ has been shown to reduce the incidence of BER during soilless tomato production. This occurs directly to the antagonistic effect between the cations in the growing medium, so that by reducing the $K^+$ concentration, the absorption and mobility of $Ca^{2+}$ can increase [67,68]. It has also been demonstrated that the use of organic fertilizers reduced the incidence of BER [69,70]. The authors found that organic fertilizers not only acted as nutrient sources and increased crop yield, but reduced the effect of BER, probably because they improved

$Ca^{2+}$ absorption and translocation. Ronga and co-authors [70] also suggested that since one of the organic fertilizers they used had milled rice bran with high levels of abscisic acid (ABA), it was possible that the surplus of ABA increased fruit $Ca^{2+}$ uptake directly, as previously reported in tomato fruits [50].

Using pericarp discs from tomato fruits [71], it was shown that exogenously applied $Ca^{2+}$ inhibited BER symptom development in a concentration-dependent manner, but increased symptom severity in tomato fruits when $Ca^{2+}$ was applied to whole plants in the irrigation solution [71]. Unexpectedly, increasing the $Ca^{2+}$ levels of tomato fruits through the expression of the vacuolar $H^+/Ca^{2+}$ antiporter, cation exchanger 1 (CAX1), from *Arabidopsis thaliana*, dramatically increased the occurrence of BER. These latter results suggest that altered $Ca^{2+}$ homeostasis between cytosolic, apoplastic, and vacuolar $Ca^{2+}$ pools might disrupt calcium signaling and lead to localized cell death and enhanced BER incidence [72]. In romaine lettuce cultivars grown in greenhouse conditions, foliar applications of $Ca^{2+}$ resulted in a significant decrease in TB symptoms, which correlated with increased $Ca^{2+}$ concentration in their young leaves as compared with non-treated controls [73]. Several authors have suggested that pectin methylesterases (PME) might be involved in $Ca^{2+}$ transport in tomato plants [74]. They found that silencing PME reduced the concentration of $Ca^{2+}$ bound to the cell wall and improved fruit tolerance to BER [74]. The overexpression of PME was shown to result in $Ca^{2+}$ translocation into cell membranes and, consequently, to $Ca^{2+}$ deficiency in most plant organs, thus enhancing BER incidence in the fruits. Other studies have suggested that the increase in PME synthesis and PME activity overlapped with the critical period for BER development [20]. Taken together, these results indicate that tightly regulated $Ca^{2+}$ homeostasis during periods of rapid growth is required to minimize BER and TB incidence in tomato and lettuce, respectively.

Two stages are involved in fruit growth: cell division influenced by auxin signaling, and cell expansion which is synergistically regulated by auxins and gibberellins (GAs). Fruit ripening occurs when auxin and GA levels decrease with a continuous increase in ABA and ethylene [75]. Phytohormones also regulate a plethora of plant responses to cope with abiotic stress factors [76–78]. Some of these hormones, such as ABA or GAs, have a direct influence on BER [18]. However, a mild level of stress, resulting from one or more interacting environmental factors, does not always result in a certain degree of BER [18]. Rather, it appears that rapid fruit growth promotes a high predisposition to BER and subsequent critical stress is required to trigger cell death [26].

Nevertheless, while it is possible that certain stress conditions may produce hormonal imbalances, it may be likely that hormones involved in cell expansion and fruit development have indirect effects on the incidence of BER. The highest concentration of auxins and GAs in the fruit occurs before cell expansion [79]. The application of auxins and/or GAs is known to increase cell division, rapid fruit growth and BER incidence [80,81]. Therefore, the acceleration of fruit growth and the inability of the plant to supply sufficient $Ca^{2+}$ to the fast-growing fruit could explain the effects of auxins and GAs on BER incidence in most cases.

Although several studies have suggested possible processes by which ABA and GAs regulate BER development in fruit tissue, many of the molecular components involved remain unknown [82]. GAs and ABA can control the expression of genes and gene networks leading to independent and/or antagonistic responses that influence fruit susceptibility to BER [83]. In tomato plants treated with GAs, the expression of genes involved in $Ca^{2+}$ transport and consequently, the concentration of water-soluble $Ca^{2+}$, was reduced and the incidence of BER concomitantly increased [84]. In turn, the addition of an inhibitor of GA biosynthesis reduced BER in fruits due to increased membrane resistance, thereby decreasing the entry of reactive oxygen (ROS) and other toxic compounds into the fruit [74].

ABA is the main hormone involved in plant stress response. Wang et al. observed that ABA levels negatively correlated with $Ca^{2+}$, suggesting that ABA plays a regulatory role in response to TB in *Brassica rapa* L. ssp. *pekinensis* [85]. Evidence has been provided indicating an antagonistic interaction between GAs and ABA in the coordination of cation

exchange activity (e.g., CAX1) in the tonoplast and thus in the incidence of BER [46,82]. The tomato *procera* (*pro*) mutant, which shows a constitutive GA response, showed a higher BER incidence due to a combined lower $Ca^{2+}$ translocation to the fruit and a reduced delivery of water and nutrients to the fruit, as a result of competition between vegetative organs and fruits for the available $Ca^{2+}$ [86].

Ethylene has also been proposed to be involved in the induction of BER [18]. Ethylene, in addition to its effect on fruit ripening, is known to be involved in the initiation of wound and pathogen responses via $Ca^{2+}$ signals [87]. Early ethylene production, premature ripening, necrosis, and cell death in the apical region of the fruit, have also been found to be symptoms directly related to BER [20,26,88]. However, it is also possible that ethylene and other stress factors that increase ROS production may influence BER, subsequent $Ca^{2+}$ concentration increase, and rapid cell expansion [88]. In persimmon fruits, salinity stress increased ethylene production, which resulted in necrotic lesions in the calyx resembling BER, but the link with endogenous $Ca^{2+}$ levels has not yet been established [89].

### 3.3. Genetic Factors Influencing BER and TB Incidence

Crop yields are strongly affected by abiotic stress caused by drought, salinity, and high temperatures. Plants respond to these stressors through various biochemical and physiological adaptations, some of which are the result of changes in gene expression [90]. In addition, many studies have emphasized that susceptibility to BER and TB is highly genotype-dependent [19,91]. In tomato, for example, pear tomatoes are more susceptible to BER than round tomatoes, and BER is never observed in cherry tomatoes [19]. In addition, a strong variation in the incidence of TB between different lettuce cultivars has been reported [92,93] that has been used for the development of TB resistant varieties through targeted breeding [92,94,95].

The use of genomic tools has allowed the identification of quantitative trait loci (QTL) for TB incidence in various recombinant inbred line (RIL) populations of lettuce and the subsequent development of linked molecular markers [96]. A major QTL accounts for up to 70% of the phenotypic variance for TB incidence in lettuce. By comparing lines with contrasting haplotypes, the genetic region was narrowed down to a genomic region containing 12 genes, two of which encoded proteins with sequence similarity to $Ca^{2+}$ transporters. These studies will allow the development of molecular markers to introgress the major resistance alleles found into new cultivars of TB-sensitive iceberg genotypes [96,97]. However, more research is needed to identify the underlying candidate genes for these QTL and to assess the effect of their introgression in other lettuce cultivars. Conversely, only a few studies have been conducted on the incidence of TB in hydroponically grown lettuce [98,99].

$Ca^{2+}$ deficiency in maize causes leaf tip rot, which is similar to TB in lettuce. Two maize lines, B73 and Mo17, differed in their $Ca^{2+}$ deficiency symptoms. In a recent study by Wang and coauthors [100], it was suggested that ammonium reduced the seedling's ability to absorb $Ca^{2+}$, which ultimately caused the observed $Ca^{2+}$ deficiency phenotype in the leaf tip. To identify a QTL associated with $Ca^{2+}$ deficiency in maize leaves, the authors used a RIL mapping population of 276 lines derived from a cross between B73 and Mo17 maize genotypes. Five QTL associated with a variation in the $Ca^{2+}$ deficiency trait were identified, and some candidate genes were selected for further studies [100].

The slow growth rate and the high concentration of $Ca^{2+}$ observed in the fruits of the IL8–3 line, which contain a small chromosome segment of the wild relative *Solanum pennellii* in the tomato cultivar M82, could be related to the low incidence of BER observed in the IL8–3 line [101]. The results of this study suggest that the main factors contributing to the difference in BER incidence between M82 and IL8–3 were fruit growth rate and $Ca^{2+}$ availability (but also other elements, including $K^+$ and $B^+$) during the early stages of fruit enlargement [102].

In a recent systematic review published by Kuronuma and Watanabe [84], the authors discussed the latest studies aimed at the identification of genes associated with BER and TB

by QTL and transcriptomic analysis. Despite these recent advances, the causative genes for $Ca^{2+}$ deficiency disorders in most crops are not yet known and await further investigation.

## 4. Solutions to Alleviate Ca-Related Disorders in Soilless Production Systems

In the present section, we will briefly introduce farmers to the tools available to minimize some physiological disorders, such as BER and TB, the incidence of which is likely to increase in the coming decades due to climate change. In intensive production systems, new strategies must be applied to mitigate these Ca-related disorders, in order to synergize crop and environmental factors to achieve efficient production with higher yields [15].

The use of smart management practices could help mitigate these Ca-related disorders but could also be useful in lessening the impact of climate change on crop productivity through better nutrient management [15,103,104]. Continuous monitoring of soilless production systems using low-cost sensors, as well as data-integration management approaches, will be key for establishing criteria and aiding decision-making during crop production [105]. It is now possible to automate a hydroponic growing system using cheap sensors that monitor and control environmental parameters such as light intensity, relative humidity, as well as pH, electrical conductivity, and temperature of the nutrient solution [106–108]. Hasan et al. [109] used drones to detect diseases in tomatoes by analyzing foliar images, which allowed them to adjust the treatment to the most affected regions of the crop. These technologies are based on the need to apply artificial intelligence techniques, such as machine learning, that requires training the initial model with a large amount of data and then using the information gathered from the crops to make predictions [109]. Indeed, the use of sensors that measure physiological processes such as photosynthesis, transpiration, and leaf stomatal conductance, has made it possible to detect and quantify the impact of drought stress in tomato plants [104]. In a recent study, the continuous monitoring of tomatoes grown in an NFT soilless system was performed by combining Netatmo sensors for greenhouse microclimate data collection, with daily fertilizer usage data [110]. Based on these data and on crop yield, the authors concluded that a cost-effective and simplified smart agriculture system allows farmers to apply accurate crop production planning and decision making of cultivation activities, such as maintaining a well-balanced microclimate environment [110]. These tools allow us to remotely or automatically adjust the different abiotic factors that, as mentioned above, can trigger the appearance of BER or TB in crops. It has also been observed that regulation of the size of air bubbles in the hydroponic could increase crop yield [111]. In this sense, it has been shown that the production of microbubbles through specific injectors would facilitate the arrival of oxygen to the finest roots, which is necessary for the effective absorption of essential nutrients and plant growth [111].

New strategies have recently been studied to reduce soil contamination due to the excessive use of agrochemicals. It has been proposed that mitigating excess of plant nutrients by using nanoparticles could lead to more precise nutrition and reduced fertilization in both conventional and hydroponic cropping systems [106]. Nanoparticles have been used as slow-release fertilizers [112,113] or for the elaboration of specific biopesticides [114,115]. In this sense, nanoparticles may provide nutrients in a more soluble and available form to plants [116], and some studies have also found that the use of carbon nanotubes as a soil amendment can double tomato yields and increase agricultural production under certain conditions. Strategies such as the use of nanoparticles for fertilization could help deliver nutrients very precisely, especially at different physiological stages, and thus avoid the effects caused by nutrient imbalances in certain phases of the plant growth cycle, which are more sensitive to the appearance of BER and TB. However, it is not yet clear how the soil ecosystem may be affected by such practices, and therefore, a thorough investigation of the impact and assessment of toxicity at all levels of the ecosystem is required [117].

Plant growth-promoting rhizobacteria (PGPRs) have been used in hydroponic growing systems as biofertilizers and/or biocontrol agents with variable success [106,109,118].

Tomato plants treated with potassium-releasing PGPRs showed a greater reduction in BER levels than untreated plants, which ultimately increased yield in terms of fruit size and weight [119]. In another study, tomato plants treated with *Pseudomonas* sp. LSW25R showed a 61% reduction in BER incidence in a hydroponic system, possibly due to increased $Ca^{2+}$ uptake in their roots [120]. In addition, the exogenous application of ABA to tomato crops has been shown to reduce BER incidence at different $Ca^{2+}$ concentrations in the nutrient solution [23,46]. Additionally, the foliar application of $Ca^{2+}$ in lettuce was found to significantly reduce TB incidence [73] Taken together, the implementation of these strategies could enhance crop production and reduce the excess use of fertilizers [121].

From a genetics point of view, identifying genotypes with a high resilience to nutritional disorders, especially $Ca^{2+}$, and introgressing the causative genes through breeding, may alleviate physiological disorders such as BER and TB [84]. Targeted breeding combined with the application of precision tools in soilless cultivation will provide us with higher yields, especially in terms of fruit quality in the case of tomato [94], as well as of leaf and head quality in the case of lettuce [92,93].

### 5. Conclusions

The present review summarizes the factors and mechanisms that trigger TB and BER, and this knowledge can be used for the development of new strategies that could help us mitigate these Ca-related physiological disorders. On the one hand, this evidence can be used to develop new cultivars that are highly tolerant to the factors that cause BER and TB. On the other hand, we propose that soilless cultivation offers many advantages over conventional cultivation, as it allows for the detailed monitoring of physiological processes and nutritional balance of plants using remote sensors. The proposed multidisciplinary strategy to reduce BER and TB levels will bring us higher yields and better quality of the final product.

**Author Contributions:** Conceptualization, V.B., J.R.A.-M. and J.M.P.-P.; writing—original draft preparation, V.B. and J.R.A.-M.; writing—review and editing, J.M.P.-P.; supervision, J.R.A.-M. and J.M.P.-P. All authors have read and agreed to the published version of the manuscript.

**Funding:** This research received no external funding.

**Institutional Review Board Statement:** Not applicable.

**Informed Consent Statement:** Not applicable.

**Acknowledgments:** We are grateful to Jorge Benítez Vega (Bayer CropScience) for his support and Mario Fon for his help with English editing.

**Conflicts of Interest:** The authors declare no conflict of interest.

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
