# Peer review of "Mitigation of Calcium-Related Disorders in Soilless Production Systems"

_agronomy, doi:10.3390/agronomy12030644_

Round 1
Reviewer 1 Report
Dear Authors
Thanks for the time and efforts that you have spent to collect, arrange and present these interesting published data about climate change and its impact on crops growth, development and productivity. Below are few comments for your consideration:
1- Plagiarism is about 20% which is higher than expected for a Q1 journal
2- Please use only related references to your title
3- For the hydroponic system please use adequate and accurate photos/pictures/figures
4- I suggest you change the title to "Mitigation of calcium disorders in soilless production systems", because it is known that soilless agriculture will be safely conducted under protected agriculture systems, so climate change will not affect Ca disorders
5- you can use the reference below at abiotic stresses section
- Alam, M.S.; Tester, M.; Fiene, G.; Mousa, M.A.A.2021. Early Growth Stage Characterization and the Biochemical Responses for Salinity Stress in Tomato. Plants, 10: 1-20. https://doi.org/10.3390/plants10040712
- Sattar, F.A.; Hamooh, B.T.; Wellman, G.; Ali, M..A.; Shah, S.H.; Anwar, Y.; Mousa, M.A.A.2021. Growth and Biochemical Responses of Potato Cultivars under In Vitro Lithium Chloride and Mannitol Simulated Salinity and Drought Stress. Plants, 10: 1-12. https://doi.org/10.3390/plants10050924
Finally, thanks again for your effort in gathering all these data together in one review article which will help scientists in the field to improve their research.
Author Response
Thank you very much for your useful suggestions. We have reviewed the ms. according to your remarks. Please, see attached PDF for the point-by-point response.

Reviewer 2 Report
- Are BER and TB always the same result for all highly industrial crops when there is nutritional imbalance? It is not clear why there is special consideration on BER and TB whereas there are a lot of Ca-induced diseases in plants.
- There are some sentences without a period "." for termination. Fix all.
- It is not clear what is the objective of this paper. The authors should explicitly state it at the end of section 1.
- Why calcium disorders was chosen considering that this is not a primary macronutrient? What is the significance?
- Include also the calcium-induced stress range for higher plants and agricultural crops (deficient and toxic levels) and inorganic/organic ways of water amendments in mitigating it.
- Italicize all scientific names like Pseudomonas sp. Fix all.
- Expound the potential of using nanoparticles/nanofertilizers in this issue.
- Expound the potential of using applied computational intelligence in this issue.
- How can closed-loop aquaponics mitigate this? Only aeroponics and hydroponics have been discussed and given special attention. It would be better if you discuss aquaponics also at this is the trend now especially in urban areas with limited space.
- Indicate also the impact of using microbubbles and larger bubble sizes in relation to sedimentation of nutrient particles in the ground level of the fertigation/water chamber.
Author Response

(The authors gave the same response as above.)

Round 2
Reviewer 1 Report
Dear Authors
Thanks a lot for your efforts to revise your manuscript.
with best of luck